# Chip-Based Spectrofluorimetric Determination of Iodine in a Multi-Syringe Flow Platform with and without In-Line Digestion—Application to Salt, Pharmaceuticals, and Algae Samples

**DOI:** 10.3390/molecules27041325

**Published:** 2022-02-16

**Authors:** Joana L. A. Miranda, Raquel B. R. Mesquita, Edwin Palacio, José M. Estela, Víctor Cerdà, António O. S. S. Rangel

**Affiliations:** 1CBQF—Centro de Biotecnologia e Química Fina—Laboratório Associado, Escola Superior de Biotecnologia, Universidade Católica Portuguesa, Rua Diogo Botelho 1327, 4169-005 Porto, Portugal; jmiranda@ucp.pt (J.L.A.M.); rmesquita@ucp.pt (R.B.R.M.); 2Department of Chemistry, University of the Balearic Islands, 07122 Palma de Mallorca, Spain; edwin.palacio@uib.es (E.P.); josemanuel.estela@uib.es (J.M.E.); 3Sciware Systems S.L., 07193 Bunyola, Spain; victorcerdamartin@gmail.com

**Keywords:** iodine, spectrofluorimetry, chip-based manifold, multi-syringe flow system, in-line UV digestion, photooxidation, Sandell–Kolthoff reaction

## Abstract

In this work, a flow-based spectrofluorimetric method for iodine determination was developed. The system consisted of a miniaturized chip-based flow manifold for solutions handling and with integrated spectrofluorimetric detection. A multi-syringe module was used as a liquid driver. Iodide was quantified from its catalytic effect on the redox reaction between Ce(IV) and As(III), based on the Sandell–Kolthoff reaction. The method was applied for the determination of iodine in salt, pharmaceuticals, supplement pills, and seaweed samples without off-line pre-treatment. An in-line oxidation process, aided by UV radiation, was implemented to analyse some samples (supplement pills and seaweed samples) to eliminate interferences and release iodine from organo-iodine compounds. This feature, combined with the fluorometric reaction, makes this method simpler, faster, and more sensitive than the classic approach of the Sandell–Kolthoff reaction. The method allowed iodine to be determined within a range of 0.20–4.0 µmol L^−1^, with or without the in-line UV digestion, with a limit of detection of 0.028 µmol L^−1^ and 0.025 µmol L^−1^, respectively.

## 1. Introduction

Iodine is abundant in the oceans, as iodide, being present in the aquatic environment of the Earth, but rare in most parts of the terrestrial environment, which leads to iodine deficiency in animals and plants grown in these soils and, consequently, in populations in such areas [1,2].

Iodine intake is critical for nervous system function throughout life, but particularly during foetus development, as it is required for thyroid hormone synthesis [3]. Even in less severe iodine deficiency, a normal thyroid gland can adapt and keep thyroid hormone production within the normal range. Prolonging thyroid hyperactivity associated with such adaptation leads to thyroid growth [2].

In Europe, two-thirds of the countries reported inadequate iodine intakes, with iodine deficiency as a major public health concern as it seems to be re-emerging [4,5]. In Portugal, results point to an inadequate iodine intake in pregnant women [6], who are recommended to have iodine supplementation during pregnancy with 150–200 μg/day [7]. 

In the human diet, main sources of iodide are marine foodstuffs—fish, shellfish, algae, and sea salt [8]. Salt iodization programmes have been implemented in more than 120 countries around the world [9,10]. It is a highly cost-effective strategy to prevent iodine deficiency [10]. Despite a major global expansion of salt iodization over the past four decades, much of Europe has remained iodine-deficient [11]. 

A significant number of consumers in the world prefer natural products over artificial ones. An alternative to iodized salt is the use of naturally iodized salt (marine salt) and seaweed as an iodine source [12]. In Portugal and Spain, marine salt, by itself, contains a naturally high amount of iodine. Even though there is a high quantity of iodine in marine salt, the use of supplements is still required as an effective alternative to combat iodine insufficiency [8].

In many Asian countries, edible seaweed products are consumed as they are low in calories and full of nutrients [13]. The use of iodine-rich seaweed for consumption is a method largely used by Japan, being the only population in the world with an excessive intake of iodine [2]. Iodine is accumulated from seawater into seaweed, making it a good dietary source of iodine. Disorders resulting from iodine deficiency can be eliminated with the adequate consumption of seaweed [13]. 

To assess iodine intake, different methods can be used to quantify iodine affordably and accurately in soil, plants, various foods, and physiological samples [14]. Iodine measurement is carried out mostly by a kinetic spectrophotometric method, the Sandell–Kolthoff reaction. The reaction is based on the reduction of yellow Ce(IV) by As(III) to colourless Ce(III), which is very slow. This reaction is catalysed by trace amounts of iodide. These procedures can be executed manually or could be automated [14,15]. To a smaller extent, the reaction is also catalysed by iodate, in the presence of arsenite, which is readily converted to iodide in an acidic medium. Various organic substances can potentially interfere in the Sandell–Kolthoff reaction, by chelating Ce(IV) or Ce(III) or directly affecting the reaction rate. Organo-iodine compounds will not react without a previous decomposition. Therefore, if the total content of iodine is the aim, complete sample mineralization is required to convert the iodine-organic forms to what can be considered iodine free from organic matter [14]. 

Several pre-treatment methods can be applied for the decomposition of the organo-iodine compounds. The pre-ashing temperature procedure can be used but may result in a significant loss of the analyte. Nevertheless, this is the method of choice for sample preparation for iodine determination by the US Food and Drug Administration. The digestion step usually uses perchloric acid, which requires special hoods and precautions. Acid digestion procedures do not have this problem. However, iodide in acid solution is easily oxidized by air, though studies show that iodine loss is consistent and is typically below 20% [14]. Ammonium persulphate was proposed as an alternative to replace chloric acid as the oxidising reagent [16]. The ammonium persulphate reagent is a non-explosive and less hazardous chemical, preventing the need for the use of a specialized hood. Applying this oxidising agent to eliminate interfering substances in urine became the method of choice in many laboratories, as the results of this method and the chloric acid one correlated very closely [15,17].

The potentiometric method is a reference method for iodide determination [15,18]. Another method of choice is the titration method, which is an accurate, easy to operate, and low-cost method that can be used for iodide determination in iodised salt samples [8,19]. There are several other analytical methods for iodine quantification, including semi-quantitative methods, the microplate method, automated methods, and technologically advanced methods, such as the inductively coupled plasma mass spectrometry (ICP-MS) method [15]. The latter allows an excellent sensitivity and, in some cases, allows the sample to be directly analysed after dilution. However, only a part of the iodine present is ionized (~25%), and so it is necessary to have an internal standard to account for matrix effects. The matrix effects need to be corrected for, for example, the salt content of a sample, as the extent of ionization in the plasma is susceptible to the ionizable material present [14]. More sophisticated and automated technology can be used, for example, paired-ion reversed-phase high-performance liquid chromatography (HPLC), but is associated with a higher cost of the instrumentation [15]. 

Concerning flow-based systems, several methods for iodine determination are described in Appendix A.

Among these, a portable, robust, and simple method for in-field analysis of iodide in sea waters samples was proposed by Frizzarin et al. [20]. It was composed of a miniaturized analyser, including a poly(methyl methacrylate) chip with integrated spectrofluorimetric detection. The solutions were propelled by a multi-syringe module. Iodide was determined through its catalytic effect on the reaction between Ce(IV) and As(III). The spectrofluorimetric detection method makes it a more sensitive method than spectrophotometric detection. The chip manifold allows for direct detection and solutions mixture in one device. The use of a multi-syringe module allows the manipulation and the quantity of reagents used to be reduced, enabling the automation of the Sandell–Kolthoff classic reaction. 

In this work, an adaptation of the above-mentioned method is proposed for the determination of iodine in pharmaceutical and salt samples. Additionally, aiming to assess the total iodine content in algae-based supplements, an oxidation process is proposed. By incorporating an in-line digestion process with UV radiation, it is possible to analyse samples with high expected levels of organo-iodine compounds. By combining an in-line digestion process with fluorometric detection, an approach that is simpler, faster, and more sensitive than the classic approach of the Sandell–Kolthoff reaction was developed.

## 2. Materials and Methods

### 2.1. Reagents and Solutions

All the solutions were prepared with analytical grade chemicals and Milli-Q water (MQW) (resistivity > 18 MΩ cm, Millipore, Bedford, MA, USA).

A 7.9 mmol L^−1^ (1.0 g L^−1^) iodide solution was prepared from the 0.10 mol L^−1^ iodide stock solution (sodium iodide) acquired from Hanna instruments (HI 4011-01, Hanna Instruments, Woonsocket, RI, USA). The working solutions were prepared daily, within a range of 0.20–4.0 µmol L^−1^ (0.20, 0.40, 0.80, 2.0, and 4.0 µmol L^−1^) of iodide.

A cerium solution containing 1.85 mmol L^−1^ Ce(IV) and an arsenious solution containing 100 mmol L^−1^ As(III) and 0.43 mol L^−1^ NaCl (Ce(IV) and As(III) solutions) were prepared in 1 mol L^−1^ H_2_SO_4_ from appropriate amounts of ammonium cerium(IV) sulphate dihydrate (Sigma-Aldrich, Steinheim, Germany), sodium (meta)arsenite (Sigma-Aldrich, Germany), and sodium chloride (Merck, Darmstadt, Germany) and a sulphuric acid stock solution, respectively.

The sulphuric acid solution, 1 mol L^−1^, was prepared by dilution of the concentrated acid (d = 1.84, 95–97%, Fluka, Taufkirchen, Germany) in MQW.

An oxidant solution, 0.3% of potassium peroxodisulfate, was prepared by dissolving 0.30 g of potassium peroxodisulfate (Merck, Germany) in 100 mL of 1 mol L^−1^ of H_2_SO_4_.

### 2.2. Chip-Based and Multi-Syringe Flow System Manifold and Procedure

#### 2.2.1. Iodine Determination

A chip (6.5 mm long, 4.4 mm wide, and 1.4 mm height) was constructed in poly(methyl methacrylate) (PMMA) with a 3D printer (Form1 +, Formlabs, Somerville, MA, USA) by using the Rhinoceros software (Appendix A). The chip includes a helicoidal microflow-channel (1.2 mm, 520 mm long) and connections, for the reagents and carrier solutions, inserted in the chip in a confluence mode. The spectrofluorimetric detection was integrated in the structure of the chip, a 2 mm optical path flow cell with an approximately 8 µL internal volume. The two optical fibres (600 µm core) were inserted in the chip, perpendicularly positioned, to irradiate the solution in the flow path from a 25 W deuterium source (Ocean Optics DH-2000-BAL, Dunedin, FL, USA). The emitted radiation is transmitted to a CCD multichannel spectrometer (Ocean Optics HR4000, Largo, FL, USA). The Autoanalysis software was employed for the system control and SpectraSuite for data acquisition. The fluorescence emission registration was set to 365 nm, subtracting the registered baseline signal, at 285 nm. The integration time was set to 300 ms for all spectrofluorimetric measurements. The PTFE tubes (0.8 mm i.d.) were used for the multi-syringe-chip connections and holding coils.

A multi-syringe system was based on 3 glass syringes (5 mL each) (Figure 1) with solenoid valves (V_i_) placed on the head section of the syringes, allowing solutions handling. Another solenoid valve (V_S_) was added to perform sample injection.

The system manifold used for the determination of iodine is described in Figure 1. Each syringe Si has a solenoid valve (Vi) which can be connected to the reservoirs of reagent or the manifold and the solenoid valve, V_S_, can be connected to the sample or to the manifold. The system routine was operated according to the protocol described in Table 1.

Firstly, the syringes were filled with the reagents solution and carrier (MQW) (step 1). Next, 2.15 mL was dispensed to allow for the aspiration of the sample (step 2). Then, the sample solution was aspirated to fill the connecting tube (step 3). For cleaning the analytical path (step 4), the carrier was propelled through the holding coil (HC), the connecting tube (L), and the chip with the activation of S_3_. Before initiating the loop for the iodine determination, all the syringes were refilled with reagent and carrier solutions (step 5).

The sample solution was aspirated to HC (step 6) and then a small sample volume (0.05 mL) was propelled to the L tube (step 7). Afterwards, reagents and sample solutions were simultaneously introduced in the chip (step 8) and propelled with the carrier solution in a lower flow rate for signal acquisition (CCD) with a 2 mm flow cell placed at the end of the chip (step 9). In the end, the analytical path was cleaned (step 10) and the syringes were refilled with reagents and carrier solutions for the next determination (step 11), repeating the determination loop for each sample a total of 3 times.

#### 2.2.2. Total Iodine Determination with In-Line UV Digestion

To attain in-line digestion, the analytical manifold was reconfigured to accommodate an additional syringe (S_4_) with an oxidant reagent (potassium peroxodisulfate 0.3% solution), resulting in another layout (Figure 2). In this method for total iodine determination, the mixture of the sample with the oxidant reagent was accomplished with the addition of a confluence (Y) after the solenoid valve (V_S_). A heated UV digester (Global FIA, Fox Island, WA, USA) set at 75 °C (D) was placed after the confluence, to promote the digestion. A debubbler device (DB) was added to prevent air bubbles from entering the chip.

The system routine was operated according to the protocol described in Table 2. 

The preparation steps, namely, filling the syringes and sample channel preparation, are analogous to those previously described in Table 1 (steps 1 to 5).

The determination of total iodine also started with sample aspiration to the holding coil (step 6). Then, the sample was mixed at the confluence (Y) with the oxidant reagent (step 7). This mixture was propelled through the UV light unit (D) to promote organo-iodine compounds digestion and the elimination of interferents (photooxidation) (step 8).

Afterwards, the reagent solutions were introduced into the chip with the digested sample and propelled from the UV light unit (D) through the debubbler (DB) to promote their mixture (step 9). Signal acquisition (CCD) was attained while propelling the mixture with the carrier solution at a lower flow rate through a 2 mm flow cell placed at the end of the chip (step 10). In the end, the analytical path was cleaned (step 11) and the syringes were refilled with reagents and carrier solutions for the next determination (step 12).

### 2.3. Sample Collection and Preparation

The two approaches developed, with and without in-line sample digestion, targeted the analysis of two types of samples: iodine supplements (dried algae and seaweed) and pharmaceuticals and food salt, respectively.

#### 2.3.1. Salt Samples

A total of 13 culinary use sea salt samples were purchased from local supermarkets. The salt samples were dissolved in MQW, 8 g of salt in 25 mL, and then diluted to 1/40, also in MQW. To determine the iodine concentration in the salt samples, calibration curves were performed using the MS-Chip method without in-line digestion.

#### 2.3.2. Iodine Supplement Samples

Two types of iodine supplements were analysed, pharmaceutical samples and dried algae pills. The supplements were purchased at a local pharmacy and an herbal shop, respectively. The suspension of the pharmaceutical and algae pills samples was made as described in Table 3.

The preparation of the #Pharm 4, #Pharm 5, and #Algae 1 sample solutions consisted of directly dissolving the pills in 100 mL of MQW and then diluting them to 1/10, also in MQW. 

The sample solutions of #Pharm 1, #Pharm 2, #Pharm 3, #Algae 2, and #Algae 3 were obtained by weighting 1/10 of the pill mass and dissolving it in 100 mL of MQW.

Additionally, an edible algae sample (#Algae 4) was also analysed; the sample solution was obtained by weighting 1 g of dried seaweed, soaking it in 100 mL of MQW, and heating it to 37 °C for 30 min.

The iodine concentration in the pharmaceutical samples was calculated by performing calibration curves using the MS-Chip method without in-line digestion.

To accomplish the iodine determination in algae samples, algae supplements pills, and seaweed, calibration curves were determined using the MS-Chip with in-line UV digestion.

### 2.4. Accuracy Assessment

#### 2.4.1. Determination of Iodine

To validate the method, several salt samples were analysed using a potentiometric method. For iodide determination, an anion selective electrode (iodide electrode (HI 4111, Hanna Instruments, Woonsocket, RI, USA)) was used and, for iodate determination, an iodometric titration method was chosen [8].

The use of the selective iodide electrode and the titration method, as reference methods, allowed the value for the inorganic iodine (iodide plus iodate value) content in the salt samples to be assessed and compared with the MS-Chip developed method.

Two pharmaceutical samples were analysed with the developed MS-Chip method for iodide determination and the result was compared with the expected values.

#### 2.4.2. Determination of Total Iodine Using an In-Line UV Digestion

Several algae supplement pills were analysed using the developed total iodine determination method (MS-Chip in-line UV digestion method) and the results compared with the supplement label value. The validation was made through recovery percentage with the algae supplement and a seaweed sample.

## 3. Results and Discussion

Two procedures were developed in a chip-based approach: one method allowed the determination of the iodine content without pre-treatment, suitable for the salt and pharmaceutical samples; the other method included an in-line digestion step combining a UV digestor and oxidant reagent, being suitable for the algae supplement samples. 

Several parameters were set according to a previous work [20], namely, the emission wavelength, reagent concentration, sample volume, and flow rates. To minimize the influence of the Schlieren effect [21], the registered base line signal at 285 nm was subtracted to the analytical signal registered at 365 nm.

### 3.1. Study of Fluorometric Determination of Iodine 

To achieve a method with a wider range of concentration for iodide determination, some parameters and conditions were revisited and studied. Maintaining the reagents and sample volume conditions reported in Frizzarin et al. [20], the influence on the Ce(IV) reagent concentration was studied in range of 1 mmol L^−1^ to 5 mol L^−1^ of Ce(IV). The Ce(IV) concentration 1.85 mmol L^−1^ was chosen, as it was the concentration that showed the highest sensitivity (Figure 3A). 

The concentration of the As(III) reagent solution was increased from 60 mmol L^−1^ to 100 mmol L^−1^ of As(III) and 0.43 mol L^−1^ of NaCl was added to this reagent solution. This alteration was based on the results described in Machado et al. (2017) [17], where the conditions of the Sandell–Kolthoff reaction method were improved. This alteration resulted in an improvement of 19% in the sensitivity.

The influence of the sulphuric acid concentration was studied for the reagent solutions preparation. A range of 137 mmol L^−1^ to 2 mol L^−1^ of H_2_SO_4_ was tested for both reagents (Ce(IV) and As(III)). The sulphuric acid concentration chosen was 1 mol L^−1^ of H_2_SO_4_, as it was the concentration that showed the highest combined sensitivity (Figure 3B).

It has also been reported that the presence of chloride added to the arsenic reagent helps the conversion of iodate to iodide in less than 1 min [22]. In order to study if the reagent conditions in the developed method, described previously, allowed the conversion of iodate to iodide, the analysis of an iodate and iodide standard was performed.

A standard with 0.99 µmol L^−1^ of iodate and 0.99 µmol L^−1^ of iodide, to a final concentration of 1.98 µmol L^−1^ of inorganic iodine, was injected and the result compared to that of a standard of 1.98 µmol L^−1^ of iodide. There was no significant difference (−2%) between the combined iodide and iodate standard, and the iodide standard of 1.98 µmol L^−1^. This indicated that the iodate present in the standard solution is converted to iodide and the total inorganic iodine can be quantified with the described MS-Chip system. 

### 3.2. Study of the Fluorometric Determination of Total Iodine

To allow a mixture of an oxidant reagent on the way to the digester, a fourth syringe and a confluence were added to the developed system (Figure 2). A fourth syringe was filled with 1 mol L^−1^ of H_2_SO_4_ to study the dilution effect of the confluence addition in the total iodine determination system. These changes reduced the sensitivity to half, as was expected. The amount of standard or sample injected was set to half (150 µL) and mixed with 150 µL of the oxidant reagent, contained in the fourth syringe. In the end, the same volume of 300 µL was propelled to the chip to be mixed with the Sandell–Kolthoff reaction reagents (Ce(IV) and As(III)).

Aiming to study the determination of the total iodine, an oxidant reagent, 0.15% of a potassium persulfate, was used based on a procedure developed by Santos et al. (2013) [23].

#### Study of the Temperature Influence

To study the conversion of the diverse iodine forms to iodide, three temperatures—25, 45, and 75 °C—were tested. Iodoacetic acid standards were used as a model to study the efficiency of the conversion of organo-iodine compounds to iodide [24,25]. Calibration curves with iodoacetic acid and iodide standards were compared. Higher temperatures were not studied because of the air bubble formation. Nevertheless, a debubbler device (Figure 2, DB) was added before the chip, to eliminate the formed bubbles.

At 25 °C, the conversion was not sufficient, about 30%, even using a higher concentration of potassium persulfate (0.30%). Therefore, the reactor temperature was increased to 45 °C. In these conditions, there was an efficient conversion of the diverse iodine forms to total iodide, as there was no statistical difference between the calibration curve with iodoacetic acid and iodide standards (Figure 4A). 

A study with the temperature up to 75 °C was carried out and an efficient conversion of the diverse iodine forms to total iodide was attained together with an improvement of 45% in the sensitivity (Figure 4A). Aiming to improve the determination sensitivity, the temperature of 75 °C was chosen.

### 3.3. Study of the Flow Rate Influence Using the MS-Chip In-Line UV Digestion System

In order to increase the contact time between the reagents and sample, and the digestion reaction extension, the influence of the flow rate of the developed method was studied.

The results with the initial flow rate of 10 mL min^−1^ for both aspiration and propulsion were compared, using calibration curves with iodide standards, to a 5 mL min^−1^ flow rate. There was no significant difference between results with the two flow rates (Figure 4B).

Aiming to decrease the time per determination, flow rates of 5 mL min^−1^ for propulsion and of 10 mL min^−1^ for the aspiration were tested (Figure 4B). 

There was an increase of sensitivity of 35% using the combined flow rate (aspiration 10 mL min^−1^ and propulsion 5 mL min^−1^) when compared with the flow rate of 5 mL min^−1^. In comparison with the flow rate of 10 mL min^−1^, the use of a combined flow rate of 10 and 5 mL min^−1^ demonstrated an increase of 25% in sensitivity.

In conclusion, using the aspiration flow rate of 10 mL min^−1^ and the propulsion rate of 5 mL min^−1^ was the best combination, as it allowed the time per determination to be decreased and the digestion reaction extension to be increased (Figure 4B).

### 3.4. Interference Assessment

#### 3.4.1. Interferences of the Sandell–Kolthoff Reaction

As thiocyanate and ascorbic acid are known to interfere in the Sandell–Kolthoff reaction [14,22,26,27], causing a signal increase with a consequent overestimation of iodine concentration, an interference study for these species was conducted. For this study, the conditions previously established, namely a temperature of 75 °C and 0.15% of the oxidant reagent concentration, were used and the interference percentage calculated.

In the study conditions, the tested thiocyanate concentrations did not interfere (signal variation < 9%) in the determination (Table 4).

For the elimination of the ascorbic acid interference, three concentrations of the oxidant potassium persulfate were tested (0.15, 0.22, and 0.3%). The interference of the ascorbic acid was eliminated (signal variation < 4%) when a concentration of potassium persulfate of 0.3% was used (Table 4).

As a conclusion of this study, an option was made to set the persulfate concentration to 0.3% for further studies. As this change could potentially influence the sensitivity of the determination, a study was conducted by tracing calibration curves with different potassium persulfate concentrations (Figure 5A).

There was no statistical difference (9%) when the potassium persulfate concentration was increased from 0.15% to 0.3% (Figure 5A). The increase of the oxidant concentration to 1.2 and 3.0% led to a decrease in the sensitivity (−45 and −56%, respectively).

To confirm that, in these modified conditions, the conversion efficiency was still maintained, a study was carried out using three different concentrations (0.15, 0.22, and 0.3%) of potassium persulfate; calibration curves with standards with iodoacetic acid and iodide were traced (Figure 5B).

It was observed that the conversion efficiency was not still statistically different from 100%. In the developed method, 0.3% of potassium persulfate solution was set as the oxidant reagent concentration.

#### 3.4.2. Other Potential Interfering Ions

To evaluate the interference of other anions in the determination of total iodine, an interference study was carried out for several anions usually present in algae [28,29,30,31]. The tested ions and the respective interference percentages are presented in Table 5.

The results demonstrated the maximum amount that did not interfere with the iodine determination using the MS-Chip in-line UV digestion method.

### 3.5. Figures of Merit

The characteristics of the two developed methods for iodine determination are summarized in Table 6.

The limits of detection and quantification, LOD and LOQ, were calculated according to IUPAC recommendations [32,33]: three (LOD) and ten (LOQ) times the standard deviation of 10 consecutive injections of MQW were used.

To perform one analytical curve with eight concentration values in triplicates, 24 analytical cycles were needed. As for an individual sample in triplicate, only three analytical cycles (without time variation) were needed. An analytical cycle was the sum of the time needed for each step.

The consumption values of effluent production per calibration curve of eight standards and reagents consumption per determination were also calculated.

### 3.6. Application to Iodine Containing Samples—Accuracy Assessment

#### 3.6.1. Salt Samples

Accuracy assessment for inorganic iodine was obtained by analysing a total of 13 marine salt samples with the developed MS-Chip method and with the comparison method for iodide and iodate determination, namely, potentiometric detection and an iodometric titration method. A linear relationship between the MS-Chip method ([Iodine]_MS-Chip_) and comparison method ([Iodine]_CM_) was established (Appendix A): [Iodine]_MS-Chip_ = 0.9956 (±0.0473) × [Iodine]_CM_ − 1.8737 (±2.8239), where the values in parentheses are 95% confidence limits [34]. These figures show that the estimated slope and intercept do not differ statistically from values of 1 and 0, respectively. Therefore, there is no evidence for systematic differences between the two sets of results.

#### 3.6.2. Supplement Iodine Samples

Five pharmaceutical samples and one algae supplement sample were analysed with the developed MS-Chip and MS-Chip in-line UV digestion methods as described in Table 7. The iodine concentrations for the reference samples with the developed methods were compared to the expected reference values. The corresponding relative deviation for each sample was calculated.

No significant differences (relative deviation, RD < 5%) were observed for the calculated relative deviation (RD) for each analysed sample. The average of the relative deviation was 2%. A statistical test (*t*-test) was used to evaluate if the mean expected/determination value significantly differed from 100%. For a 95% significance level, the calculated t-value was 0.493 with a corresponding critical value of 3.495; the calculated t-value was lower than the critical value, thus indicating that the results are not statistically different.

### 3.7. Recovery Studies

To further validate the method, standard additions were performed, and recovery percentages were calculated to validate the MS-Chip in-line UV digestion method. A total of 0.79 μmol L^−1^ and/or 1.58 μmol L^−1^ of iodide was added to three different algae samples, and the samples were analysed by the developed method. 

The information about the different samples, the initial concentration, the added value of iodide, the concentration found, and the recovery percentages were calculated according to the IUPAC [35]. The results are summarized in Table 8.

The average of the recovery percentages was 103% with a standard deviation of 11%. A statistical test (*t*-test) was used to evaluate if the mean recovery value significantly differed from 100%. For a 95% significance level, the calculated t-value was 0.181 with a correspondent critical value of 4.177, indicating that no multiplicative interferences were found.

## 4. Conclusions

The use of a multi-syringe flow system, with the possibility of an additional in-line UV digestion step for the determination of total iodine, was demonstrated to be advantageous compared to the classic approach of the Sandell–Kolthoff reaction [17]. In the developed method, no pre-treatment steps were required, reagents and sample volume consumption were reduced, and automation was accomplished. The microfluidic approach attained with the used PMMA chip was successful as the device proved to be highly robust with no leaking problems throughout the entire work and endured the use of highly acidic reagents.

The in-line digestion approach here described presents an efficient conversion of organo-iodide compounds to iodide, avoiding any off-line treatments, which is a significant advantage over classical methods or other previously described flow methods [20,22,36,37,38]. However, both approaches cannot run simultaneously within the same manifold, requiring a physical reconfiguration. This drawback does not shadow a major advantage, the wider iodine determination range (0.20–4.0 µmol L^−1^), achieved when compared with a previous work [20] using the same fluorimetric reaction. In fact, the attained quantification range allows samples to be analysed within the reference range values of iodine intake: levels of insufficient intake within <0.16 to 0.78 µmol L^−1^, adequate levels of 0.79 to 1.57 µmol L^−1^, above requirement levels 1.58 to 2.36 µmol L^−1^, and excessive levels ≥2.36 µmol L^−1^ of iodine [39].

Additionally, the developed method can be effectively applied to iodine determination in salt, algae supplements, seaweed, and pharmaceutical samples, which are examples of intake forms of iodine in the population.

## Figures and Tables

**Figure 1 molecules-27-01325-f001:**
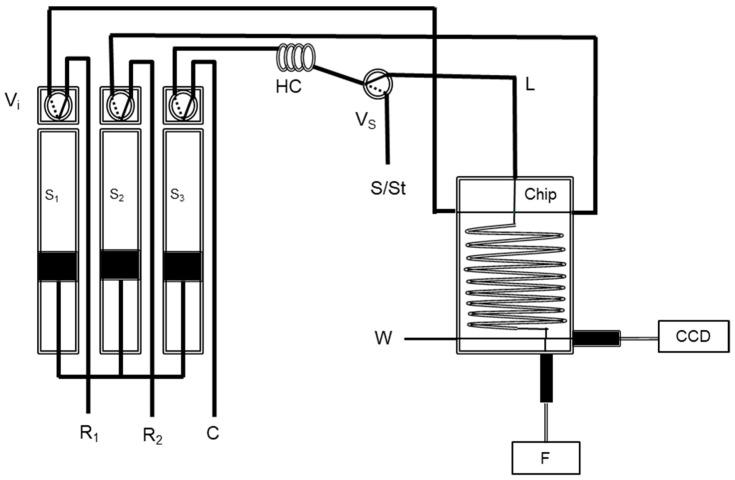
Flow diagram of the chip multi-syringe flow injection analysis for iodine determination: S_1_–S_3_, syringe pumps; V_i_, two-way solenoid valves for each syringe pump; Vs, two-way solenoid valve for sample insertion; R_1_, 1.85 mmol L^−1^ Ce(IV); R_2_, 100 mmol L^−1^ As(III), both R_i_ in 1 mol L^−1^ H_2_SO_4_; C, carrier (MQW); S/St, sample/standard; HC, 200 cm holding coil: L, tube length, 20 cm; F, irradiation from a UV-VIS-NIR light source Micropack DH 2000-BAL; CCD, charged coupled device detector connected with optical fibre (600 µm core) for fluorescence measurement (Ocean Optics HR4000); W, waste.

**Figure 2 molecules-27-01325-f002:**
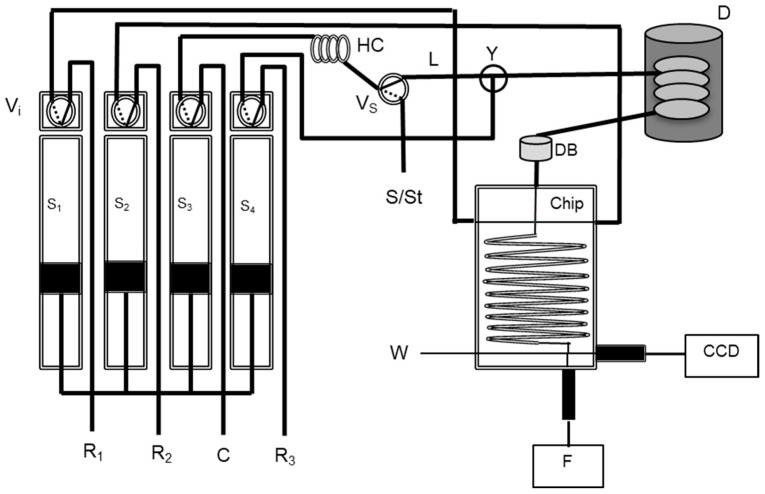
Flow diagram of the chip multi-syringe flow injection analysis for total iodine determination: S_1_–S_4_, syringe pumps; V_i_, two-way solenoid valves for each syringe pump; Vs, two-way solenoid valve for sample insertion; R_1_, 1.85 mmol L^−1^ Ce(IV); R_2_, 100 mmol L^−1^ As(III); R_3_, 0.30% potassium persulphate, all Ri in 1 mol L^−1^ H_2_SO_4_; C, carrier (MQW); S/St, sample/standard; HC, 200 cm holding coil; L tube length, 2 cm; D, heated UV digester (Global FIA) with 1000 mL holding coil; DB, debubbler; Y, confluence and length to chip 1640 cm; F, irradiation from a UV-VIS-NIR light source Micropack DH 2000-BAL; CCD, charged coupled device detector for fluorescence measurement (Ocean Optics HR4000) connected through an optical fibre (600 µm core); W, waste.

**Figure 3 molecules-27-01325-f003:**
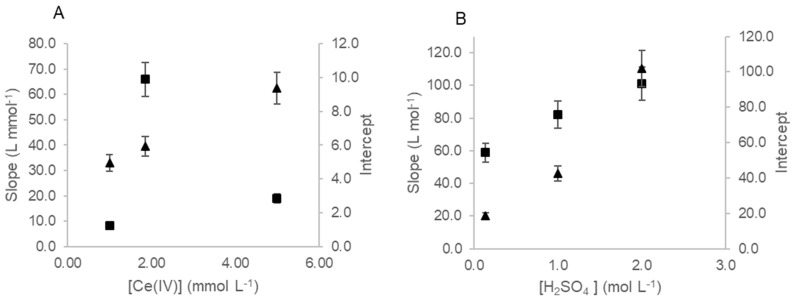
Study of the influence of the reagents solution concentration: (**A**) influence of Ce(IV) reagent solution concentration on the analytical curve slope (■) and intercept (▲); (**B**) influence of H_2_SO_4_ concentration in the 1.85 mmol L^−1^ of Ce(IV) reagent solution on the analytical curve slope (■) and intercept (▲).

**Figure 4 molecules-27-01325-f004:**
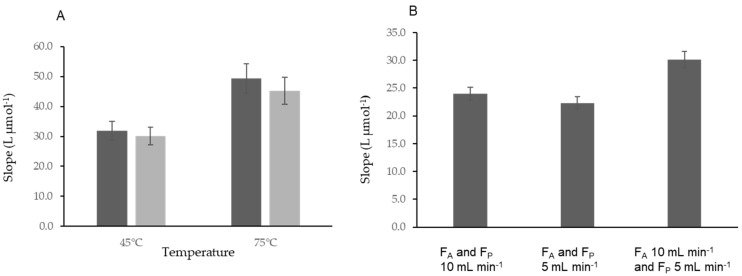
Study of the MS-Chip in-line UV digestion system. (**A**) Effect of the temperature on the conversion of diverse iodine forms to iodide. Dark grey bars, iodoacetic acid standards; light grey bars, iodide standards. (**B**) Influence of the flow rate with the MS-Chip in-line UV digestion system. F_A_, aspiration flow rate; F_P_, propulsion flow rate.

**Figure 5 molecules-27-01325-f005:**
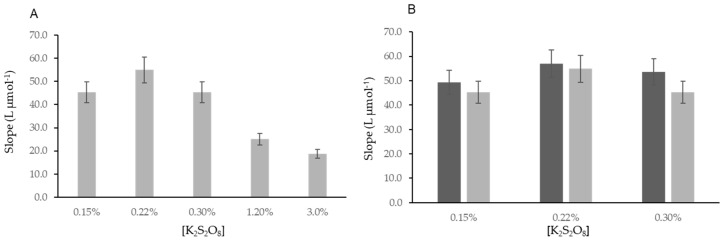
Study of the influence of different oxidant reagent concentrations (**A**) in the calibration curve with iodide standards and (**B**) in the conversion of different iodine forms into iodide. Dark grey bars, iodoacetic acid standards; light grey bars, iodide standards.

**Table 1 molecules-27-01325-t001:** Protocol for the spectrofluorimetric determination of iodine, using a multi-syringe chip-based flow system (MS-Chip method).

	Step	Active Devices	Action	Flow Rate	Description
Preparation Steps	1	S_1_, S_2_, S_3_	Aspirate 5.00 mL	10 mL min^−1^	Filling of syringes with reagents and carrier
	2	S_1_, S_2_, S_3_	Dispense 2.15 mL	5 mL min^−1^	Enabling the syringes to have capacity for sample aspiration
	3	S_3_, V_S_	Aspirate 0.150 mL	10 mL min^−1^	Sample aspiration for channel preparation
	4	S_3_	Dispense 3.00 mL	5 mL min^−1^	Cleaning the analytical path
	5	S_1_, S_2_, S_3_	Aspirate 1.50 mL	10 mL min^−1^	Filling of syringes with reagents and carrier
Loop for iodine determination	6	S_3_, V_S_	Aspirate 0.35 mL	10 mL min^−1^	Sample aspiration
	7	S_3_,	Dispense 0.05 mL	5 mL min^−1^	Sample injection
	8	S_1_, S_2_, S_3_	Dispense 0.30 mL	5 mL min^−1^	Reagents and sample injection into the chip
	9	S_3_	Dispense 0.90 mL	0.4 mL min^−1^	Transport through the chip and signal acquisition (CCD)
	10	S_3_	Dispense 0.60 mL	5 mL min^−1^	Cleaning the analytical path
	11	S_1_, S_2_, S_3_	Aspirate 1.50 mL	10 mL min^−1^	Refilling of syringes with reagents and carrier

**Table 2 molecules-27-01325-t002:** Protocol for the spectrofluorimetric determination of total iodine, using a multi-syringe chip-based flow system with in-line UV digestion (MS-Chip in-line UV digestion method).

	Step	Active Devices	Action	Flow Rate	Description
Preparation Steps	1	S_1_, S_2_, S_3_, S_4_	Aspirate 5.00 mL	10 mL min^−1^	Filling of syringes with reagents and carrier
	2	S_1_, S_2_, S_3_, S_4_	Dispense 2.35 mL	5 mL min^−1^	Enabling the syringes to have capacity for sample aspiration
	3	S_3_, V_S_	Aspirate 0.150 mL	10 mL min^−1^	Sample aspiration for channel preparation
	4	S_3_	Dispense 3.00 mL	5 mL min^−1^	Cleaning the analytical path
	5	S_1_, S_2_, S_3_, S_4_	Aspirate 3.32 mL	10 mL min^−1^	Filling of syringes with reagents and carrier
Loop for Total Iodine Determination	6	S_3_, V_S_	Aspirate 0.150 mL	10 mL min^−1^	Sample aspiration
	7	S_3_, S_4_	Dispense 0.150 mL	5 mL min^−1^	Sample injection and oxidant reagent propeled through Y confluence for mixing
	8	S_3_	Dispense 1.02 mL		Propeling the sample and oxidant reagent mixture through the UV digestor
	9	S_1_, S_2_, S_3_	Dispense 0.300 mL		Reagents and sample injection into the chip
	10	S_3_	Dispense 0.900 mL	0.4 mL min^−1^	Transport through the chip manifold and signal acquisition (CCD)
	11	S_3_	Dispense 1.10 mL	5 mL min^−1^	Cleaning the analytical path
	12	S_1_, S_2_, S_3_, S_4_	Aspirate 3.32 mL	10 mL min^−1^	Refilling of syringes with reagents and carrier

**Table 3 molecules-27-01325-t003:** Preparation steps of the samples to be analysed in the MS-Chip flow system method.

	Sample ID	Amount of Sample (Dissolved in 100 mL in MQW)	Dilution Factor
Pharmaceuticals	#Pharm 1	1/10 of pill	-
	#Pharm 2		
	#Pharm 3		
	#Pharm 4	Pill	10×
	#Pharm 5		
Algae Pills	#Algae 1	Pill	10×
	#Algae 2	1/10 of pill	-
	#Algae 3		

**Table 4 molecules-27-01325-t004:** Assessment of the Sandell–Kolthoff reaction interferents with the developed MS-Chip in-line UV digestion method, using an iodide standard of 100.0 µg L^−1^ and 0.3% of potassium persulfate as oxidant reagent.

Interferent	Concentration (mg L^−1^)	Interference (%)
SCN^−^	5	3
	22	9
C_6_H_8_O_6_	6.6	6
	13.2	4
	26.5	13

**Table 5 molecules-27-01325-t005:** Assessment of the influence of potential interfering ions present in algae with the developed MS-Chip in-line UV digestion method, using an iodide standard of 200.0 µg L^−1^.

Potential Interferent	Reference Concentration in Algae (mg Kg^−1^)	Expected Concentration in Tested Sample (mg L^−1^)	Tested Concentration (mg L^−1^)	Interference (%)
NO_3_^−^	4500 **	45	50	−4
			150	−10
			500	−20
NO_2_^−^	40.9 **	0.41	40	−7
			50	−11
PO_4_^3−^	5400	54	10,000	−2
			15,000	−11
CN^−^ *	0.3	0.003	2.2	3
			9.9	9
C_6_H_8_O_6_ *	118.8	1.2	13.2	4
			26.5	13

* Using an iodide standard of 100 µg L^−1^. ** Reference concentration in plants.

**Table 6 molecules-27-01325-t006:** Features of the two developed MS-Chip methods for iodine quantification in salt and algae samples.

	Dynamic Range (µmol L^−1^)	Typical Calibration Curve ^a^ A = S × µmol L^−1^ I + b	LOD (µmol L^−1^)	LOQ (µmol L^−1^)	One Determination (h^−1^)	Analysis Rate (h^−1^) ^b^	Effluent Production (mL) ^b^	Reagent Consumption (µmol) ^c^
MS-Chip	0.20–4.0	A = 91.17 ± 1.24 × [I] + 65.87 ± 9.326 R^2^ = 0.9985 ± 0.0016	0.025	0.199	0.049	1.18	71	(Ce(IV)) 0.555 (As(III)) 30.0 (K_2_S_2_O_8_) 1.65 ^d^
MS-Chip w/ in-line UV digestion	0.23–4.0	A = 42.18 ± 3.524 × [I] + 177.7 ± 15.86 R^2^ = 0.9963 ± 0.0010	0.028	0.231	0.058	1.40	110	

^a^ Five calibration curves. ^b^ Calibration curve with eight standards. ^c^ One determination. ^d^ Oxidation reagent.

**Table 7 molecules-27-01325-t007:** Results obtained with the proposed flow system, MS-Chip method without and with in-line UV digestion, for five pharmaceutical reference samples (#Pharm) and one algae supplement sample (#Algae). RD, relative deviation; SD, standard deviation.

Sample ID	Expected Value [I^−^] (µmol L^−1^)	MS-Chip Method [I^−^] ± SD (µmol L^−1^)	RD (%)
#Pharm 1	0.81	0.83 ± 0.02	1.5
#Pharm 2	1.76	1.78 ± 0.01	1.5
#Pharm 3	2.52	2.61 ± 0.20	3.6
#Pharm 4	1.58	1.60 ± 0.01	1.3
#Pharm 5	2.36	2.40 ± 0.03	1.5
#Algae 1	1.58	1.66 * ± 0.20	5.2

* MS-Chip in-line UV digestion method.

**Table 8 molecules-27-01325-t008:** Recovery percentages calculated from spiked supplement algae and seaweed samples assessed with the MS-Chip in-line UV digester developed method. SD, standard deviation; RSD, relative standard deviation.

Sample ID	Initial		Added	Found		Recovery (%)
	[I] ± SD (µmol L^−1^)	RSD (%)	[I] (µmol L^−1^)	[I] ± SD (µmol L^−1^)	RSD (%)	
#Algae 2	1.74 ± 0.01	0.6	1.58	3.33 ± 0.03	0.9	100
#Algae 3	2.13 ± 0.05	2.3	0.79	2.97 ± 0.17	5.7	106
			1.58	3.96 ± 0.15	3.8	116
#Algae 4	0.27 ± 0.02	7.4	0.79	0.97 ± 0.12	12	89

## Data Availability

Data are contained within the article or Appendix A.

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
