# Peer review of "Chip-Based Spectrofluorimetric Determination of Iodine in a Multi-Syringe Flow Platform with and without In-Line Digestion—Application to Salt, Pharmaceuticals, and Algae Samples"

_molecules, 2022, doi:10.3390/molecules27041325_

Round 1

Reviewer 1 Report

The present manuscript entitled "Chip-based spectrofluorimetric determination of iodine in a multi-syringe flow platform with and without in-line digestion – application to salt and algae samples" by Joana L. A. Miranda, Raquel B. R. Mesquita, Edwin Palacio, José M. Estela, Víctor Cerdà, and António O. S. S. Rangel (molecules-1600964) describes the development of a flow-based spectrofluorimetric method for iodine determination. The applied flow manifold consists of a 3D printed miniaturized chip, a multi-syringe module, and a liquid driver. This system was reconfigured to provide in-line digestion. Analytical performance was checked based on the analysis of salt, algae, seaweeds, and pharmaceutical samples.

The present article is very well written and has a good structure. The article is interesting from an analytical point of view; therefore, it should interest the journal's reader. I would like to congratulate the authors on a perfect publication. The paper meets Molecules' requirements, and I recommend the article for publication in Molecules following the common editing stage. My current decision is a minor revision. More specific comments and observations are presented below.

  1. Title. The authors also presented pharmaceutical analyzes in the publication; therefore, pharmaceutical samples can be added to the title.
  2. Materials and Methods. It would be good to give the exact concentrations, not just ranges.
  3. Unit of concentration. I have the impression that the "L-1" in the unit is written with a font of a different size.
  4. When specifying the instruments, please remember to add the country of origin, not just the company, e.g., in the 3D printer and CCD multichannel spectrometer.
  5. Page 3, line 148. What was the guiding principle when choosing this wavelength?
  6. Description of Fig. 1 and Fig. 2. There is no explanation for V1.
  7. Table 1 and Table 2. The "Active devices" column is missing an entry in some places. Were no devices active in these steps?
  8. Page 5, lines between 180 and 187. In the text, you can add shortcuts to given elements of the flow system in appropriate places (Y, S4, R3, DB, D).
  9. Section 2.3. Could I ask for a more detailed description of the samples tested? It would be valuable for the publication. Were all dilutions performed with MQW?
  10. The Authors presented a study on temperature and flow rate. Why exactly were such parameter values chosen? I would also like to ask if experimental planning methods were considered?
  11. Section 3.4.1. What is the type of interference effect?
  12. Page 14, line 446. Are you sure that this is a multiplicative effect?
  13. Does the developed method have disadvantages?
  14. Conclusion. Please emphasize clearly the advantages of the research carried out in relation to [20].
  15. Supplement Material. Authors in [33] and [38] do not have to be in capital letters.

I hope that the comments presented will help improve the article.

Reviewer 2 Report

Comments to the Authors:

Manuscript Number: 1600964

Title: Chip-based spectrofluorimetric determination of iodine in a multi-syringe flow platform with and without in-line digestion – application to salt and algae samples

The manuscript is correctly written, scientifically sound, and clearly presented in good written English. Employed experimental methods are adequate, clear, and complete to allow repetition of the work. Data are properly interpreted to support the conclusions. Relevant issues in the discussion are adequately discussed.

This is an interesting manuscript introducing novel manufacturing of multi-syringe flow microfluidic device using a 3D printer.

Major concerns:

It would be advisable to include other polymer materials e.g. cyclo-olefin copolymer (COC), cyclic olefin polymers (COP) in the research as well. Also, the author should elaborate on why PMMA was chosen as a microchip material.

I suggest testing the effect of pH on the behavior of microchip material.

Have the authors tested the system for leakage? If so how was the testing conducted?
